# Suppressive Antibiotic Therapy in Prosthetic Joint Infections: A Contemporary Overview

**DOI:** 10.3390/antibiotics14030277

**Published:** 2025-03-07

**Authors:** Hajer Harrabi, Eloïse Meyer, Nathalie Dournon, Frédérique Bouchand, Christel Mamona Kilu, Véronique Perronne, Karim Jaffal, Emma d’Anglejan, Clara Duran, Aurélien Dinh

**Affiliations:** Infectious Disease Department, University Hospital R. Poincaré, Assistance Publique des Hôpitaux de Paris, Versailles Paris Saclay University, 104 Bd R. Poincaré, 92380 Garches, France; eloise.meyer@aphp.fr (E.M.); frederique.bouchand@aphp.fr (F.B.); christel.mamonakilu@aphp.fr (C.M.K.); veronique.perronne@aphp.fr (V.P.); karim.jaffal@aphp.fr (K.J.); emma.danglejanchatillon@aphp.fr (E.d.); clara.duran@aphp.fr (C.D.)

**Keywords:** SAT, suppressive, surgery, PJI

## Abstract

The management of prosthetic joint infections (PJIs) poses significant challenges, requiring a multidisciplinary approach involving surgical, microbiological, and pharmacological expertise. Suppressive antibiotic therapy (SAT) has emerged as a viable option in cases where curative interventions are deemed unfeasible. This review provides an updated synthesis of recent evidence on SAT, including its indications, efficacy, practical considerations, and associated challenges. We aim to highlight the nuances of this therapeutic approach, discuss the factors influencing its success, and offer future directions for research to optimize patient outcomes.

## 1. Introduction

The management of prosthetic joint infections (PJIs) is a complex process that requires the integration of surgical, microbiological, and pharmacological considerations, all of which must be tailored to each individual patient to achieve the best possible outcome.

The primary objective in treating PJIs is to eradicate the infection while preserving or restoring joint functionality. While surgical options such as debridement, antibiotics, and implant retention (DAIR), or one- and two-stage revisions, remain mainstays, some clinical scenarios contraindicate their use [1,2].

Indeed, in some cases, curative treatment of the infection may not be feasible, while in others, the likelihood of surgical success is deemed very low. As a result, the prolonged use of antibiotics to suppress the infection’s progression becomes a valuable option. This strategy is called suppressive antibiotic therapy (SAT) [3,4].

Historically, SAT for PJI has been linked to poor outcomes. However, success rates over the past decade have varied between 60% and 93% [4]. Suppressive antibiotic therapy has gained traction as a strategy to control infection and alleviate symptoms when curative surgery is impractical or has a high likelihood of failure. The prolonged use of antimicrobials to suppress infection without achieving eradication represents a paradigm shift in PJI management [5].

## 2. Concept and Definition of SAT

SAT entails long-term or indefinite administration of antibiotics aimed at mitigating symptoms and delaying disease progression. It is typically employed when surgical intervention is not feasible due to patient-related factors or when curative strategies have limited success potential [1]. In the area of PJI, SAT is considered a “noncurative” strategy [3]. A significant challenge in treating infections associated with prosthetic materials is the ability of many bacteria to form biofilms. These bacterial biofilms are microscopic matrices that adhere to both living tissue and prosthetic surfaces [4,6], often forming rapidly after infection. This impairs achieving a complete cure, as these biofilms protect bacteria from both antimicrobial agents and the host immune system [7]. Bacteria in biofilms on prosthetic materials can evade detection by standard microscopy and culture techniques [4]. The use of dithiothreitol (DTT) as an antibiofilm pretreatment in fluid samples has shown potential in improving microbiological yield and diagnostic sensitivity by disrupting biofilm structures without killing bacteria. Integrating such strategies into routine diagnostics could enhance the detection of biofilm-related infections, warranting further investigation and clinical implementation [8]. Biofilms necessitate suppressive strategies to manage persistent infections and prevent their escalation [6,9].

However, there is no global consensus on what constitutes SAT, with definitions varying significantly across studies and clinical settings. In most European publications, SAT refers to lifelong antibiotic treatment for ‘incurable infections,’ typically in patients who receive suboptimal or no surgical intervention [10]. Conversely, studies from the United States primarily use the term SAT to describe extended antibiotic treatment following DAIR. These two concepts of SAT represent distinct treatment strategies with differing goals and durations:
Fixed-term SAT: Prolonged antimicrobial therapy for a defined duration of 6–24 months, with the primary aim of curing the infection (American approach).Indefinite SAT: Antimicrobial therapy for an undetermined duration, intended to prevent relapse (European approach).

In both approaches, SAT is initiated after the infection has been clinically controlled in accordance with standard treatments outlined by national, international, or local guidelines [10]. This variability complicates efforts to standardize treatment and assess outcomes effectively.

## 3. Indications for SAT, Dosage, Duration

### 3.1. Indications

SAT is primarily indicated in:Acute PJIs: Particularly when DAIR fails or has limited likelihood of success [11].Chronic PJIs: Where resection arthroplasty or revision surgery is not an option due to high surgical risk, short life expectancy, or other contraindications [12].Failed prior treatments: Cases involving recurrent infection or unsuccessful curative attempts [13].

Despite limited literature supporting its utility, SAT is frequently employed following DAIR in clinical practice, even by experienced Orthopedic Infectious Diseases specialists. Decisions to initiate and continue SAT should be guided by a comprehensive risk-benefit assessment that considers multiple factors. These include the perceived risk of failure based on patient and infection characteristics known to predict treatment outcomes, the feasibility of further surgical intervention given anatomic and implant conditions, the patient’s willingness to undergo additional surgery if necessary, and the potential adverse effects of prolonged antimicrobial therapy [14].

However, beyond the concept of DAIR, a new approach has recently emerged: the “augmented DAIR” concept, more commonly known as DAPRI [15].

Interestingly, regional practices vary widely. An online survey investigating clinicians’ approaches to SAT for PJI revealed that North American physicians are significantly more likely to prescribe SAT for acute PJIs following DAIR compared to their European counterparts (38% versus 6%) [10]. For patients with PJI not managed surgically, the majority of respondents in this study indicated that SAT is appropriate, provided the patient does not have a fistula. The host risk factors for failure that were considered indications for initiating SAT in acute PJI treated with DAIR were frail patients, megaprothesis, chemotherapy, no change of modular components during surgery, poor soft tissue, immunosuppressive drugs, a second debridement less than three weeks after DAIR, rheumatoid arthritis, and no use of rifampicin in staphylococcal PJI. The top five microbiological factors cited as indications for SAT were infection with *Candida* species, *Pseudomonas* species, rifampicin-resistant Staphylococci, methicillin-resistant *Staphylococcus aureus* (MRSA), and Enterococci [10].

In a recent review published by Cortes-Penfield et al., the authors propose risk–benefit stratification criteria to guide the use of SAT following DAIR for PJI [14].

Indications strongly supporting SAT (at least one of these factors): (a) when surgical revision is not an option (i.e., requiring amputation, arthrodesis, or complex wound reconstruction); (b) in cases of recurrent PJI or previous treatment failure; (c) infections caused by difficult-to-treat pathogens (*S. aureus*, *P. aeruginosa*, *Candida* sp.), which have higher recurrence risks, justifying prolonged suppression; (d) severe immunosuppression (solid organ/stem cell transplant, chemotherapy, chronic steroids, TNF inhibitors, advanced HIV); (e) patients who underwent arthroscopic DAIR or retained the polyethylene liner.Indications where SAT may be considered: (a) major comorbidities (cirrhosis, End-Stage Renal Disease, heart failure); (b) older patients (>75 years) or those with limited life expectancy (<10 years); (c) late hematogenous infections (onset > 2 years post-arthroplasty) with or without active bacteremia; (d) gram-negative infections untreatable with fluoroquinolones; (e) SAT may also be appropriate when patients prioritize infection suppression over surgical revision.Factors suggesting limited benefit of SAT: (a) patients who have completed six weeks of rifampin for monomicrobial coagulase-negative staphylococci PJIs, or those who have received a full fluoroquinolone-based regimen for gram-negative infections; (b) culture-negative PJIs, where targeted suppression is impossible.

These criteria are based on the limited literature reviewed and reflect their consensus practice [14]. They are designed to support decision-making by considering the patient’s individual risk of recurrent infection, the potential consequences of treatment failure, and their values and preferences. They recommend that SAT is most appropriate for patients at the highest risk of failure (e.g., those with limited surgical source control, recurrent PJI, and/or difficult-to-treat pathogens) and for those where further failure could result in catastrophic functional outcomes due to limited surgical alternatives [14]. Currently, there are no specific recommendations for SAT use in guidelines.

### 3.2. Antibiotic Dose in SAT

Infectious Diseases Society of America (IDSA) recommendations for lower doses of some antimicrobials are primarily based on expert opinion, as studies specifically evaluating SAT dosing remain limited [16]. While many physicians worldwide commonly use relatively low antibiotic doses for SAT, there is currently only one retrospective study addressing this approach [17]. This study compared patients with orthopedic implant infections treated with low-dose versus normal-dose SAT, after an initial standard treatment of 1 to 2 weeks of intravenous antibiotics followed by 4 to 11 weeks of targeted oral antimicrobial therapy (for a total duration of 6 to 12 weeks) [17]. Low dose SAT following initial treatment varied depending on antibiotics, but was approximately half of standard dose.

No significant difference in failure free survival between patients on low-dose SAT and those on standard-dose SAT was observed. Lower dose did not seem to reduce side effects. Sub-MIC levels of antibiotics affect bacterial virulence and biofilm formation, thereby causing undesirable infections. Additional research is needed to better understand the efficacy of low-dose SAT and its potential impact on the development of antimicrobial resistance.

### 3.3. Duration

Once SAT is initiated, the decision to continue or discontinue therapy should be regularly reevaluated [14]. Although earlier studies did not define a point beyond which SAT ceases to delay or prevent infection, two recent retrospective cohort studies found no difference in treatment failure rates between PJI patients receiving 1 year of oral antimicrobial suppression versus those receiving longer durations following DAIR [18,19]. This indicates that the majority of SAT’s benefit likely occurs within the initial months, and a defined-duration approach may be as effective as lifelong SAT [18,19].

Rather than considering SAT as a lifelong commitment, it is advisable to inform patients that its continuation will be reassessed after 1 year, with discontinuation being a reasonable option for most individuals at that point [14]. Regular monitoring of SAT should be maintained through outpatient visits, typically annual or biannual after an initial phase of closer follow-up. These visits should include symptom evaluation and laboratory testing to detect potential adverse reactions to antibiotics or signs of relapse or reinfection [14].

The heterogeneity of patient populations between studies makes it difficult to provide recommendations for clinical practice. There are currently no definitive guidelines to determine a clinically relevant stopping point, nor a biomarker that may indicate to safely stop SAT. Some authors have also suggested radiolabeled leukocyte scintigraphy as possible methods for monitoring response to SAT but this strategy needs to be confirmed in larger studies [20].

## 4. Efficacy of SAT

### 4.1. Does SAT Work?

Evidence supporting SAT remains heterogeneous and predominantly observational. Studies indicate variable success rates ranging from 23% to 84%, depending on the criteria used to define efficacy [21,22]. For patients treated with DAIR, SAT has been associated with reduced odds of treatment failure, with one meta-analysis reporting a 4-fold reduction in recurrence risk [22].

Recent large cohort studies have demonstrated success rates of approximately 58% to 75% within 2 years of SAT initiation, dropping to around 50% at 5 years, indicating its potential for long-term infection control [23]. The variability in outcomes highlights the importance of tailoring SAT protocols to individual patient needs and infection profiles. Notably, patients in settings with robust multidisciplinary teams (MDTs) may experience better outcomes due to coordinated care and comprehensive follow-up [14].

Interpreting the efficacy of SAT is challenging for several reasons: the lack of controlled studies, the inclusion of patients with acute infections who may be cured by DAIR alone, and inconsistencies in the criteria used to evaluate efficacy across published studies summarized in Table 1. For instance, some authors defined success as avoiding surgery, even if the infection was not fully controlled [5], whereas others also required symptom control as part of their efficacy criteria [13,22,24,25]. Reported success rates ranged from 23% to 84%, but the studies with the highest success rates predominantly included patients with early PJI [13,22,25], many of whom might have achieved similar outcomes with significantly shorter treatment durations.

### 4.2. Predictors of Success and Failure

Factors influencing SAT outcomes include:Pathogen type: Gram-positive organisms (enterococci), especially *Staphylococcus aureus*, *Candida* spp. and *Pseudomonas aerugionsa*, are associated with higher failure rates [27].Biofilm formation: Infections involving biofilm-associated organisms are less likely to respond to SAT [29].Patient demographics: Advanced age and severe comorbidities such as cirrhosis or chronic kidney disease increase the likelihood of SAT failure [30].Regional variability: Practices differ significantly, with variations in antimicrobial selection, dosing, and duration reflecting differences in regional guidelines and microbial resistance patterns [14].

Failures are often due to antibiotic-resistant organisms, inadequate adherence to prescribed regimens, or the emergence of unsuspected pathogens. Incorporating biomarkers like C-reactive protein (CRP) and erythrocyte sedimentation rate (ESR) into monitoring protocols can aid in assessing treatment response and identifying early signs of relapse [33].

Few studies have evaluated the factors associated with SAT failure. However, failure rates appear to be higher in patients with a sinus tract and in those with infections caused by *S. aureus* [21,22,33].

In a multicenter study published by Escudero-Sanchez et al., predictors of failure were analyzed, with failure defined as either persistent, uncontrolled symptoms of PJI (including a sinus tract) or the need for additional surgery (such as debridement or prosthesis removal) due to infection [35]. A multivariate analysis identified the following factors as being associated with SAT failure:Infection etiology other than Gram-positive cocci (e.g., Gram-negative rods, fungi, or negative cultures): This may be due to the limited availability of orally active antimicrobials for Gram-negative bacilli.Prosthesis located in the upper limbs: Although this finding is difficult to explain, it may be influenced by the relatively small number of upper-limb PJI cases.Age under 70 years: This seemingly paradoxical finding might reflect that younger patients managed with SAT are more likely to be immunosuppressed or have “tumoral” prostheses, which are associated with a worse prognosis [29].

At present, there are no definitive predictors of SAT failure. Therefore, SAT should not be excluded as a treatment option for patients who meet the appropriate conditions despite the factors mentioned above [3].

## 5. Practical Considerations

### 5.1. Is Debridement Necessary?

While debridement reduces bacterial load and provides microbiological samples, its necessity prior to SAT initiation is debated. Retrospective studies indicate that SAT can succeed even without debridement in patients with stable symptoms [42]. However, surgical intervention is recommended to optimize infection control when feasible [43].

### 5.2. Optimal Antibiotic Regimens

Selection of antibiotics for SAT prioritizes oral bioavailability, tolerability, and efficacy against causative pathogens. Common regimens include monotherapy with beta-lactams, tetracyclines, or combinations involving rifampin [44]. Long-acting agents like dalbavancin have emerged as promising options for patients with limited compliance [36].

Tetracyclines and cotrimoxazole are particularly favored for their tolerability and low risk of resistance development. However, care should be taken when selecting antibiotics for polymicrobial infections or gram-negative pathogens, as these cases often require combination therapy [45].

Dalbavancin is a broad-spectrum lipoglycopeptide effective against Gram-positive infections, including MRSA. Due to its spectrum, prolonged action, and favorable tolerability, dalbavancin appears promising for suppressive antibiotic, as reported in a pilot study [41].

### 5.3. Role of Initial Intravenous Therapy

Although initial intravenous therapy is commonly employed, its necessity remains unclear. Clinical guidelines recommend at least 6 weeks of intravenous antibiotics for severe cases before transitioning to oral SAT [32]. Evidence suggests that shorter intravenous durations may be sufficient for stable infections, provided oral alternatives are effective and well-tolerated [46].

### 5.4. Treatment Interruptions

Antibiotic-free intervals are generally discouraged due to high failure rates associated with treatment discontinuation, particularly within the first four months [28]. Long-term suppressive regimens ensure continuous bacterial suppression and reduce the risk of relapse. However, intermittent SAT strategies may be considered for patients experiencing significant side effects or adherence challenges [31].

## 6. Safety and Adverse Events

Prolonged antibiotic use carries risks, including gastrointestinal disturbances, skin reactions, and nephrotoxicity. Notably, *Clostridioides difficile* infections and antimicrobial resistance have been documented, necessitating careful monitoring [32]. Adverse events are reported in up to 41% of SAT patients but rarely lead to discontinuation [33].

In a multicenter cohort study, SAT was discontinued due to adverse events in only 5.6% of patients, demonstrating its overall safety when carefully managed [34]. Nevertheless, the emergence of resistant pathogens underscores the need for antimicrobial stewardship and periodic reassessment of treatment efficacy.

Emerging concerns include the potential for microbiome disruption and its long-term implications. Future research should explore strategies to mitigate these risks while preserving SAT efficacy [35].

## 7. Conclusions and Future Directions

SAT represents a viable, albeit non-curative, option for managing complex PJIs. It has shown promise in extending infection-free intervals and improving quality of life for patients ineligible for surgical interventions. However, its long-term benefits must be weighed against risks such as antimicrobial resistance and adverse events [37].

The global heterogeneity in SAT practices underscores the need for standardization. Future research should prioritize defining optimal indications, antimicrobial regimens, and monitoring protocols. Large-scale randomized controlled trials and international collaborations will be instrumental in addressing these gaps [14].

## Figures and Tables

**Table 1 antibiotics-14-00277-t001:** Published Series on SAT in PJI.

Author [Ref] Methods	Type of Infection	Previous Surgical Treatment	Bacteria Major Strains Isolated	Suppressive Antibiotics (N Patients)	Duration Mean (Mo)	Follow up Mean (Mo)	Success Rate (%)	Conclusion
Johnson and Bannister (1986) [26] Retrospective 25 cases SAT 9 cases	56% acute 44% chronic	Excision of sinus tract, debridement, exchange arthroplasty	*S. aureus* (52%), CoNS (28%), *Streptococcus* spp. (20%)	No data	15.6 (1.2–59)	15.6 (1.2–59)	8	SAT very rarely eradicate deep infection in a cemented prosthesis
Goulet et al. (1988) [5] Retrospective (1972–1982) 19 cases	90% chronic 10% acute	DAIR (11/19; 57.9%)	*S. aureus* (21%), CoNS (21%), *Streptococcus* spp. (32%)	Penicillin 8 Ampicillin 5 Cefazolin 4 Gentamicin 3 Oxacillin/Dicloxacillin 3 Clindamycin 1 Erythromycin: 1	14 patients (73.7%): without a planned endpoint 4 patients (21%) 39 (12–104) 1 patient (5.3%): 6	49.2 (24–120)	63.2	SAT is indicated in old, frail patients SAT may also be considered for an otherwise compliant patient who refuses removal of an infected prosthesis
Tsukayama et al. (1991) [23] Retrospective 13 cases	100% chronic	DAIR	*S. aureus*, (54%), CoNS (46%)	No data	No data	37.2 (24–55)	23	SAT has limited clinical efficacy SAT is associated with a substantial risk of adverse effects
Segreti et al. (1998) [25] Retrospective (1986–1992) 18 cases	50% chronic 50% acute	DAIR in all cases	*S. aureus* (44%), CoNS (44%)	Minocycline 5 Dicloxa/Oxa 5 Penicillin 2 Ampicillin 1 TMP/SMX 1 Other ATB: 4	48.9 (4–103)	48.9 (4–103)	83.3	SAT is a reasonable alternative to surgery in selected patients with infected orthopedic prostheses
Rao et al. (2003) [22] Retrospective 36 cases (1995–2001)	53% chronic 47% acute	DAIR in all cases	*S. aureus* (26%), CoNS (50%)	Minocycline/ Rifampin: 11 Levofloxacin: 5 Cephalexin: 4 Dicloxacillin: 3 Sulfamethoxazole/Trimethoprim: 2 Minocycline: 2 Oxacillin: 2 Penicillin: 2 Clindamycin: 1 Amoxicillin/Doxycycline: 1 Fluconazole: 1 Linezolid: 1	52.6 (6–128)	60 (16–128)	86.2	The ideal regimen and optimal duration of oral SAT is not well-established -Prospective studies are needed
Marculescu et al. (2006) [21] Retrospective (1995–1999) 99 episodes in 91 patients	No data	DAIR in all cases A median of 1 surgical debridement per patient (range, 1–4 debridement)	*S. aureus* (32%), CoNS (23%)	Oral b-lactam antibiotics 53% (penicillins in 17 episodes and cephalosporins in 36 episodes) Minocycline 7% Trimethoprim sulfamethoxazole 10% Quinolones: 8%	23.3 (0.33–92.6) 0.03 Mo = 1 day	23.3 (0.16–89.1)	57	The role of a sinus tract and duration of symptoms are important to predict the success of debridement and retention of prosthesis Future clinical trial studies
Byren et al. (2009) [13] Retrospective 112 cases (1998–2003)	31% chronic 69% acute	Open debridement (97; 87%) Arthroscopic washout (15; 13%) Multiple procedures (24; 21%)	*S. aureus* (40%), CoNS (23%)	FQ/RFP combinations of doxycycline, fusidic acid, rifampicin, clindamycin or amoxicillin	12 months at least	27.6	82	The length of duration of antibiotic prescribing beyond 6 months is not critical to the outcome Prospective controlled trials
Prendki et al. (2014) [9] Retrospective (2004–2011) 38 cases	61% chronic 39% acute	(9; 23.7%) Synovectomy 6/9 Abscess drainage 3/9 Partial exchange 1 Excision of fistula 1	*S. aureus* (39%), *Streptococcus* spp. (18%), GNB (17%)	Amoxicillin (8), amoxicillin–clavulanate (1), cloxacillin (4), clindamycin (7), co-trimoxazole (1), fusidic acid (5), minocycline (1), levofloxacin (1), peflacin (2), and rifampin (13)	59 (15–90)	24 (6–98)	60	SAT is an alternative therapy in elderly patients with PJI when surgery is contraindicated
Siqueira et al. (2015) [27] Retrospective (1996 to 2010) 92 cases	61% chronic 39% acute	All cases Irrigation and debridement with polyethylene exchange (54; 58.8%), 2-stage revision (38; 41.2%)	*S. aureus* (48%), CoNS (35%)	Dicloxacillin (13) Doxycycline (29) Cephalexin (8) Trimethoprim/sulfamethoxazole (12) Amoxicillin (6) Clindamycin 300 mg bid (4) RFP (2) FQ (2) Other ATB	63.5 ± 38.3 (6–165.1)	69.1 ± 38.2 (2.2–168.3)	68.5	Chronic suppression with oral antibiotics increased the infection-free prosthetic survival rate following surgical treatment
Prendki et al. (2017) [28] Retrospective cohort 21	No data	No data	*S. aureus* (62%), CoNS (21%)	- Monotherapy clindamycin (5/21), beta-lactams (4/21), co-trimoxazole (4/21), pristinamycin (4/21), and fluoroquinolones (4/21) - Dual therapy: (4/21) fluoroquinolone + rifampicin, fluoroquinolone + clindamycin, co-trimoxazole + fusidic acid, and amoxicillin + clindamycin.	12.7 (1.3–56.5)	Over follow-up 17.3 (1.3–56.6) Follow-up under SAT 9.2 (1.3–56.5)	66	SAT appeared to be an effective and safe option in this cohort
Pradier et al. (2017) [12] Retrospective cohort (2006–2014) 39 cases	61% delayed or late 39% acute	DAIR (32; 82.1%) Implant exchange (7; 17.9%)	*S. aureus* (79%), CoNS (10%)	Doxycycline	- Mean duration 22.5 ± 20.6 (17–28) Two-year duration of SAT (13, 33.3%) A continued SAT (26; 67.7%)	24	74.4	Oral doxycycline used as SAT in patients treated for *S. aureus* PJI has an acceptable tolerability and effectiveness, and appears to be a reasonable option in this setting
Wouthuyzen-Bakker et al. (2017) [29] Retrospective (2009–2015) 21 cases	62% late or delayed 38% early	A debridement and/or lavage of the affected joint (14; 67%)	*S. aureus* (33%), CoNS (38%)	Clindamycin (83%) minocycline (67%) Amoxicillin 4/21 19% Amoxicillin/acide clavulanique 2 × 21 (9.5%) Moxifloxacine 2/21 (9.5%)	No data	21 (3–81).	67	SAT is a reasonable alternative treatment option in a subgroup of patients with a PJI who are no candidate for revision surgery, in particular in patients with a ‘standard’ prosthesis and/or CoNS as the causative micro-organism
Pradier et al. (2018) [30] Retrospective (2006–2014) 78 cases	60% delayed or late 40% early	DAIR 59; 75.6% Implant exchange 19; 24.4% including 1SE (10.3%) and 2SE (11.5%) and Resection arthroplasty management (2; 2.6%)	*S. aureus* (40%), CoNS (32%)	Doxycycline (72; 93.6%) Minocycline 6; 6.7%)	22.2 ± 17.9	34 ± 19.9	71.8	Oral cyclins used as SAT in patients treated for PJI have an acceptable tolerability and effectiveness a reasonable option
Weston et al. (2018) [31] Retrospective 134 cases	Acute 100% Acute postoperative infection 17% Acute hematogenous infection 83%	DAIR 100%	*S. aureus* 29% CoNS 23%	No data	No data	60 (25.2–156)	66	The greatest risk factor for SAT failure was an infection with a staphylococcal species, followed by age of <60 years
Pouderoux et al. (2019) [32] Single-center prospective cohort study (2010–2018) 10 cases	Acute 7; 70% Chronic 3; 30%	DAIR 6; 60%	Gram negative bacilli 5; 50% Polymicrobial 4; 40% *Streptococcus* spp. 1; 10%	Ertapenem 7; 70% Ceftriaxone 2; 20% Ceftazidim 1; 10%	14.4 (IQR 6.98–23.7), for a total of _6000 subcutaneous injections.	14	60	As salvage therapy, subcutaneous SAT delivered by gravity infusion is a safe and interesting alternative when an optimal surgical strategy is not feasible and no oral treatment is available
Leijtens et al. (2019) [33] Retrospective (2006–2013) 23 cases	30% early 70% late or delayed	20; 87.5% underwent surgery before the start of AST DAIR 13; 56.5%	*S. aureus* (2%), CoNS (61%)	Doxycycline 14; 60.8% TMP/SMX 6; 26%	38 (1–151)	33	56.5	SAT an alternative treatment in selected patients with a PJI There is a persisting and considerable number of failures, particularly in PJI caused by *S. aureus* and in patient with an antibiotic-free period before the start of SAT
Renz et al. (2019) [34] Prospective cohort study (2016–2018) with a retrospective control group (2009–2015) 69 cases SAT 24 cases (35%)	Early 12; 17% Delayed 27; 39% Late 30; 43%	DAIR 27; 39% One-stage exchange in 5; 7%, Multi-stage exchange 31; 44% Prosthesis removal 6; 9%	Beta-hemolytic *Streptococci* spp. 43; 62% *S. viridans* group 26; 38%	Amoxicillin 22/24 doxycycline 1/24 Clindamycin 1/24	13 (0.5–111)	13 (0.5–111)	95	SAT was associated with higher success rate compared with no suppression (93% vs. 57%, *p* = 0.002) SAT should be strongly considered in streptococcal PJI.
Sandiford et al. (2020) [7] Retrospective (2012–2017) 24 cases	No data	DAIR 15; 62.5% Single-stage revision 4; 16.6%, two-stage revision 4; 16.6%	*S. aureus* (25%), CoNS (21%)	Amoxicillin (no data) Doxycycline (no data) Fluconazole 1 case	No data	122.8 (15.6–68.4)	83	SAT is a viable option for the management of PJI with a low incidence of complications
Escudero- Sánchez et al. (2020) [35] Retrospective multicenter cohort (2003–2016) 302 cases	73% chronic 11% hematogenous 16% early postoperative	Debridement with partial removal 24; 7.9%) Debridement without removal 143; 47.4% Non-surgical 132; 43.7%	*S. aureus* (31%), CoNS (33%)	Tetracyclin 39.7% TMP/SMX 35.4% Rifampicin in combination with another antibiotic 23.2%	36.5; IQR 20.75–59.25	36.5	58.6	SAT offers acceptable results for patients with PJI when surgical treatment is not performed or when it fails to eradicate the infection
Lensen et al. (2020) [36] Multicenter, retrospective observational cohort study (2008–2018) 72 cases SAT 63 cases	Chronic 100%	No data	Cocci G positive (70%) Gram negative bacilli (24%)	TMP/SMX 25% Fluoroquinolones 7%	No data	54.4	No data	SAT is not able to fully prevent complications in patients with a draining sinus. However, it may be beneficial in a subset of patients
Ferry et al. (2021) [37] Prospective 4 cases	Chronic 100%	LysinDAIR Several previous prosthetic knee revisions without prosthesis loosening	*S. epidemidis*	Tedizolid in the 4 cases	>12 months	>12 months	50	Exebacase has the potential to be used in patients with staphylococci PKI during arthroscopic DAIR as salvage therapy to improve the efficacy of suppressive antibiotics and to prevent major loss of function
Burr et al. (2022) [38] Retrospective (2007–2020) 45 cases	Chronic 100%	No data	*S. aureus* (62% 27/45) CoNS (17.7% 8/45) Gram Positive 9% (5/45)	Doxycycline 11; 24.4% Cephalexin 9; 20% TMP/SMX 7; 15.5 Combination with RFP 4; 9% Amoxicilline/Amox-clav: 4; 9% Clindamycin 3; 7%	50	50	67	SAT is a reasonable strategy in patients with PJI who lack or refuse further surgical treatment options
Ceccarelli et al. (2023) [39] Retrospective study 16 cases	Chronic 100%	DAIR in all cases	CoNS13 (82%) CoNS + *E. coli* 1 (6%) MRSA 2 (12%)	Minocyclin 100%	15 (6–30)	15 (6–30)	62.5	SAT can be considered as an interesting approach in patients not suitable for standard treatments of PJI Requires careful monitoring
Tai et al. (2024) [40] Multicenter retrospective Europe 2005–2016 USA 2008–2018 510 patients	Acute 100% Early acute infection (367; 62%) Late acute infection (143; 38%)	DAIR in all cases	*S. aureus* (38%) CoNS (29%) *Streptococcus* spp. (19%) Polymicrobial (31%)	Rifampicin 282 (55%) Quinolone 221 (43%)	No data	26.7 (3–136)	Overall succes rate 87.7% Succes rate SAT76.6%	SAT’s benefits might be restricted to specific groups of patients, underscoring the need for randomized controlled trials
Lafon-Desmurs et al. (2024) [41] Retrospective bicentric study (2021–2023) 15 cases 12 cases of PJI	No data	No data	*S. aureus* (20%), CoNS (33.3%) Polymicrobial (33.3%)	Dalbavancin The median number of injections received as SAT and excluding the loading dose was 4 (IQR 2–7)	The median time between two reinjections was 1.9 (IQR 1–2.7) with a maximum of 4.7 days	9.9	80%	These results support the use of dalbavancin SAT for implant-related infections

CDI: *Clostridioides difficile* infection; CoNS: coagulase-negative staphylococci, Mo: months; IQR: interquartile, GNB: Gram negative bacilli.

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
