# Peer review of "Suppressive Antibiotic Therapy in Prosthetic Joint Infections: A Contemporary Overview"

_antibiotics, 2025, doi:10.3390/antibiotics14030277_

Round 1
Reviewer 1 Report
Comments and Suggestions for Authors
The document is an interesting overview of suppressive antibiotic therapy, whose specific indications are still under discussion.
It is not a systematic review, but it takes up some articles from 2024 and highlights their importance and potential use.
However there are minor revisions to address.
First of all I found the bibliography too dated. The first two citations are from 2017 and 2003, number 5 is from 1988. Are we sure there is not a more recent article in the last 30 years regarding the eradication of the infection?
Precisely in this regard, when the authors discuss therapeutic options for implant preservation, I recommend studying this article "Debridement, antibiotics, pearls, irrigation and retention of the implant and other local strategies on periprosthetic hip joint infections" (10.23736/s2784-8469.21.04173-0), which goes beyond the concept of DAIR, introducing a "DAIR augmented", better known as DAPRI.
Likewise, when discussing biofilm on prosthetic surfaces, do not forget that recent articles and reviews have shown that Bacteria can live in biofilms floating in fluids (and chemical pre‑treatment of synovial fluid samples can improve microbiological counts in peri‑prosthetic joint infection). This is a topic you cannot ignore in 2025 when talking about infected prosthetics and an unidentified pathogen.
Lines 59-62: Specify which of the two approaches is the European one and which is the American one.
Line 98: At leat... at least?
Line 125: Write Infection Diseases Society of America before writing IDSA
Finally, the table cannot be accepted in its present form. I realize that reporting all this information is challenging, but the solution cannot be simply putting everything in a table: it is not readable, it is confusing, it contains too much information and with all these columns you are limited to putting only a few words per line.
I suggest you rethink the table using fewer columns, or make two tables. Or at least transcribe the most discursive information in prose in the text (perhaps with a bulleted list), and leave in the column only the data of immediate impact for the reader's vision.
Author Response
The document is an interesting overview of suppressive antibiotic therapy, whose specific indications are still under discussion.
It is not a systematic review, but it takes up some articles from 2024 and highlights their importance and potential use.
However there are minor revisions to address.
First of all I found the bibliography too dated. The first two citations are from 2017 and 2003, number 5 is from 1988. Are we sure there is not a more recent article in the last 30 years regarding the eradication of the infection?
We have added more recent references in our bibliography, especially a recent review by Robin Patel in the NEJM (Patel, R. Periprosthetic Joint Infection. N Engl J Med 2023, 388, 251–262, doi:10.1056/NEJMra2203477).
Precisely in this regard, when the authors discuss therapeutic options for implant preservation, I recommend studying this article "Debridement, antibiotics, pearls, irrigation and retention of the implant and other local strategies on periprosthetic hip joint infections" (10.23736/s2784-8469.21.04173-0), which goes beyond the concept of DAIR, introducing a "DAIR augmented", better known as DAPRI.
Likewise, when discussing biofilm on prosthetic surfaces, do not forget that recent articles and reviews have shown that Bacteria can live in biofilms floating in fluids (and chemical pre‑treatment of synovial fluid samples can improve microbiological counts in peri‑prosthetic joint infection). This is a topic you cannot ignore in 2025 when talking about infected prosthetics and an unidentified pathogen.
We thank the reviewer for these useful suggestions; we have added them to our manuscript.
Lines 59-62: Specify which of the two approaches is the European one and which is the American one.
Line 98: At leat... at least?
Line 125: Write Infection Diseases Society of America before writing IDSA
We modified our manuscript accordingly, thank you.
Finally, the table cannot be accepted in its present form. I realize that reporting all this information is challenging, but the solution cannot be simply putting everything in a table: it is not readable, it is confusing, it contains too much information and with all these columns you are limited to putting only a few words per line.
I suggest you rethink the table using fewer columns, or make two tables. Or at least transcribe the most discursive information in prose in the text (perhaps with a bulleted list), and leave in the column only the data of immediate impact for the reader's vision.
We thank the reviewer for these comments; we have simplified our table accordingly.
Reviewer 2 Report
Comments and Suggestions for Authors
This review aims to highlight the suppressive antibiotic therapy (SAT) in the management of prosthetic joint infections, including its indications, efficacy, practical considerations, and directions to optimize patient outcomes. SAT is a treatment method in which antibiotics are taken long-term or indefinitely. It is used when a bacterial infection may not be curable or other treatments are unsuccessful. The patient receiving suppressive antibiotic therapy often has more comorbidities, and infections often involve retained prosthetic material.
This review is clear, comprehensive and relevant to the field. It is still relevant and interested to the scientific community.
Main suggestion: In the section of “Antibiotic Dose in SAT”. Sub-MIC levels of antibiotics affect bacterial virulence and biofilm formation, thereby causing undesirable infections. The author can add Sub-MIC comments to this section.
Main problem: The tables in this review are too large and complex to read. Can the author rearrange or split it up to make it more readable?
Minors:
- Line 106: ESRD, abbreviation needs detailed at it first appearance.
- Line 125: IDSA, abbreviation needs detailed at it first appearance.
- Line 183: Is Clostridoides a typo errs of Clostridioides?
- Line 204: S. aureus needs to be italicized. Check the whole manuscript and make change for the same situation.
Author Response
This review aims to highlight the suppressive antibiotic therapy (SAT) in the management of prosthetic joint infections, including its indications, efficacy, practical considerations, and directions to optimize patient outcomes. SAT is a treatment method in which antibiotics are taken long-term or indefinitely. It is used when a bacterial infection may not be curable or other treatments are unsuccessful. The patient receiving suppressive antibiotic therapy often has more comorbidities, and infections often involve retained prosthetic material.
This review is clear, comprehensive and relevant to the field. It is still relevant and interested to the scientific community.
Main suggestion: In the section of “Antibiotic Dose in SAT”. Sub-MIC levels of antibiotics affect bacterial virulence and biofilm formation, thereby causing undesirable infections. The author can add Sub-MIC comments to this section.
We thank the reviewer for this suggestion; we have added this comment to our manuscript.
Main problem: The tables in this review are too large and complex to read. Can the author rearrange or split it up to make it more readable?
We have simplified the table for better understanding, thank you.
Minors:
- Line 106: ESRD, abbreviation needs detailed at it first appearance.
- Line 125: IDSA, abbreviation needs detailed at it first appearance.
- Line 183: Is Clostridoides a typo errs of Clostridioides?
- Line 204: S. aureus needs to be italicized. Check the whole manuscript and make change for the same situation.
We thank the reviewer for these comments; we have modified our manuscript accordingly.